# D-Tagatose-Based Product Triggers Sweet Immunity and Resistance of Grapevine to Downy Mildew, but Not to Gray Mold Disease

**DOI:** 10.3390/plants11030296

**Published:** 2022-01-23

**Authors:** Nikola Mijailovic, Nicola Richet, Sandra Villaume, Andrea Nesler, Michele Perazzolli, Essaid Aït Barka, Aziz Aziz

**Affiliations:** 1Induced Resistance and Plant Bioprotection, USC INRAE 1488, University of Reims, UFR Sciences, CEDEX 02, 51687 Reims, France; nikola.mijailovic@univ-reims.fr (N.M.); nicolas.richet@univ-reims.fr (N.R.); sandra.villaume@univ-reims.fr (S.V.); ea.barka@univ-reims.fr (E.A.B.); 2Bi-PA NV, 1840 Londerzeel, Belgium; andrea.nesler@bi-pa.com; 3Research and Innovation Centre, Fondazione Edmund Mach, Via E. Mach 1, 38098 San Michele all’Adige, Italy; michele.perazzolli@unitn.it; 4Centre Agriculture, Food and the Environment (C3A), University of Trento, Via E. Mach 1, 38098 San Michele all’Adige, Italy

**Keywords:** D-tagatose, IFP48, induced resistance, sweet immunity, sugar-enhanced defense, *Plasmopara viticola*, *Botrytis cinerea*, *Vitis vinifera*

## Abstract

The use of natural bio-based compounds becomes an eco-friendly strategy to control plant diseases. Rare sugars would be promising compounds as inducers of plant “sweet immunity”. The present study aimed to investigate the induced resistance of grapevine leaves against *Plasmopara viticola* and *Botrytis cinerea* by a rare sugar-based product (IFP48) and its active ingredient D-tagatose (TAG), in order to elucidate molecular mechanism involved in defense-related metabolic regulations before and after pathogen challenge. Data showed that spraying leaves with IFP48 and TAG lead to a significant reduction of downy mildew, but not of gray mold disease. The induced protection against *P. viticola* relies on IFP48’s and to a lesser extent TAG’s ability to potentiate the activation of salicylic acid- and jasmonic acid/ethylene-responsive genes and stilbene phytoalexin accumulation. Most of defense responses remained upregulated in IFP48-treated plants after infection with *P. viticola*, but inconsistent following challenge with *B. cinerea*. The beneficial effects of IFP48 were associated with an enhanced accumulation of tagatose inside leaf tissues compared to TAG treatment. Meanwhile, the amounts of sugars, glucose, fructose, maltose, galactose and trehalose remained unchanged or decreased in IFP48-treated leaves after *P. viticola* infection, although only a few genes involved in sugar transport and metabolism showed transcriptional regulation. This suggests a contribution of sugar homeostasis to the IFP48-induced sweet immune response and priming plants for enhanced resistance to *P. viticola*, but not to *B. cinerea*.

## 1. Introduction

The cultivated grapevine (*Vitis vinifera* L.) is highly susceptible to a large number of economically devastating diseases, including downy mildew, caused by the oomycete *Plasmopora viticola*, and gray mold, caused by the fungus *Botrytis cinerea* [1]. *P. viticola* is an obligate biotroph that invades grapevine through the stomata, forms a long-term feeding relationship and acquires nutrients via haustoria which are formed within mesophyll cells [1]. However, the necrotrophic *B. cinerea* first kills the host cell with toxins and hydrolytic enzymes, and then uses the macerating plant tissues as food [2]. These diseases are mainly controlled by chemical fungicides, which cause negative impacts on human health and environment, as well as the appearance of resistant pathogen strains [3,4,5]. The challenge for a sustainable viticulture is to reduce chemical inputs by the implementation of new innovative and eco-friendly strategies aiming at inducing plant resistance and inhibiting pathogen development. In this context, promising outcomes have been achieved using natural bio-based elicitors or resistance inducers of different nature, capable of triggering plant innate immunity [3,4,5,6,7,8,9,10,11].

Most resistance inducers characterized so far are bio-based compounds containing sugar repeat units [3,4,5,6,7,8,9,10,11]. They include microbe-associated molecular patterns (MAMPs), derived from fungi, bacteria or oomycetes such as β-glucans, oligochitosaccharides or glycolipids [6,7,8,9], and damage-associated molecular patterns (DAMPs) derived from the host cell wall, such as oligogalacturonides from pectin [10], cellodextrins from cellulose [3], or xyloglucans from hemicellulose [11]. Their ability to induce resistance in plants seems to be dependent not only on their backbone structure, but also on plant cultivars, pathogen lifestyle and environmental conditions [3,4,8]. The successfully induced plant resistance depends on the recognition of MAMPs or DAMPs, which trigger a cascade of early responses and a transcriptional reprogramming, characteristic of MAMP-triggered immunity (MTI). Phytohormones, jasmonic acid (JA), ethylene (ET) and salicylic acid (SA) are also produced to different levels depending on the perceived MAMPs/DAMPs, and are also involved in the regulation of MTI to ensure plant resistance to pathogens [12,13].

Increasing attention has been paid to the role of endogenous sugars in mediating plant immune response and counteracting pathogen attack [14,15,16,17], resulting in the concept of “sweet immunity” or “sugar-enhanced defense” [14,15]. Sugars are energy sources for both plants and pathogens for which they compete, and sugar transporters are the molecular actors involved in this competition and probably in the fate of the interaction. Sugars also function as signaling molecules to regulate defense gene expression and plant metabolism [16,17]. Another way that sugar signaling could mediate resistance is through modulation of the ability of pathogens to produce effector molecules in the host [18]. Plants also respond to the PAMP signals with induction of hexose/H^+^ symporters, that may limit apoplastic sugar accumulation, thus affecting pathogen virulence directly [19]. Plants can use monosaccharide transporters to quickly resorb sugars upon pathogen invasion [20,21]. By using hexose/H^+^ symporters, plants may also counteract the SWEET-mediated secretion and thus retrieve apoplastic sugars, causing pathogen starvation [21,22]. It has also been proposed that sugars may function as priming molecules leading to MTI in plants [15,22]. This concept of “sweet priming” predicts specific key roles to sugars in perceiving, mediating and counteracting pathogen attack [15].

In recent years, some rare monosaccharides and their derivatives have shown potential functional and ecological properties in controlling plant pathogens [23,24,25,26,27,28,29,30,31,32,33,34,35,36]. Their availability has been achieved through the implementation of new enzymatic and microbial processes, making their use effective in food and agricultural sectors [25,30]. D-tagatose, as a ketohexose, is among rare sugars found at low concentrations in many foods and generally recognized as safe by the Food and Drug Administration [30]. D-tagatose was shown to inhibit the growth of various plant pathogens, such as *Phytophthora infestans* causing late blight [24], *Oidium violae* causing powdery mildew in tomato [33], *Plasmopara viticola* (downy mildew) and *Erysiphe necator* (powdery mildew) in grapevine [32,33] and *Hyaloperonospora parasitica* (downy mildew) in Chinese cabbage [31]. It has been reported that D-tagatose acted directly on the pathogen by inhibiting essential enzymes leading to impairment of the whole metabolism [31]. In particular, D-tagatose inhibited fructose metabolizing enzymes (for e.g., fructokinase and phosphomannose isomerase) leading to inhibition of glycolysis and cell wall synthesis [31].

Rare sugars can also play a role in fulfilling “sweet immunity” processes and triggering plant defense [15]. It is speculated that after being exogenously applied, some rare sugars could penetrate into plant cells [37], alter the sucrose-to-hexose ratio, differentially influencing glucose [38], fructose [39] and sucrose signaling pathways [40], and thereby inducing plant immune responses [15,30]. Furthermore, exogenous application of rare sugars could be sensed through hexokinase-dependent or -independent pathways [41], thus producing an elicitor-like effect [42,43]. It is also likely that a coordinated interaction of sugar and hormonal pathways in plants leads to effective induction of defense responses [15]. Although the effectiveness of tagatose against some fungal and oomycete pathogens was reported [25,26,31,35], the molecular mechanisms involved in grapevine protection against downy mildew and gray mold remained to be elucidated.

In this study, we first investigated the ability of foliar applied D-tagatose-based product (IFP48) and pure tagatose (TAG) as active ingredient to protect grapevine leaves against *P. viticola* and *B. cinerea*. We then explored molecular mechanisms underpinning induced protection before and after pathogen challenge. We especially targeted the expression of defense hormonal-responsive genes and the synthesis of stilbene phytoalexins. We further investigated weather IFP48 and TAG could induce change in the expression of genes encoding sugar transporters and vacuolar invertase. The relationship between the efficiency of IFP48 and TAG and the accumulation of D-tagatose, as well as of endogenous sugars in leaf tissues was also assessed before and after *P. viticola* inoculation. We showed that IFP48 triggers sweet immune response and priming plants for enhanced resistance to *P. viticola*, but not to *B. cinerea*. The induced resistance against *P. viticola* relies on the accumulation of tagatose in leaf tissues and the potentiation of SA and JA/ET-dependent responses with the contribution of sugar homeostasis upon pathogen challenge.

## 2. Results

### 2.1. IFP48 and TAG Confer Leaf Protection against P. viticola, but Not B. cinerea

The capacity of IFP48 and TAG to protect grapevine leaves was assessed by foliar spraying of both products at 5 g/L, as an optimal concentration previously optimized in the lab conditions. Two days post treatment (dpt), leaves were detached from water- (control), IFP48- and TAG-treated plants, washed with Tween 80 at 0.001%, and infected with *P. viticola* and *B. cinerea*. Disease symptoms were monitored by quantifying sporulation of *P. viticola* at 7 days post inoculation (dpi) and measuring necrotic lesions of *B. cinerea* at 5 dpi and 14 dpi. Data showed that leaves from control plants were heavily colonized by *P. viticola*, while IFP48 and TAG reduced sporulation density by about 50 and 35%, respectively (Figure 1A). However, neither IFP48 nor TAG was able to reduce the severity of gray mold disease (Figure 1B,C). Most leaves developed the same symptoms as the control at both 5 dpi (Figure 1B) and 14 dpi (Figure 1C).

### 2.2. IFP48 and TAG Potentiate the Expression of SA- and JA/ET-Responsive Genes

The expression of defense genes responsive to SA (*PR1* and *PR2*), JA/ET (*PR3c*, lipoxygenase 9 (*LOX9)*, 1-aminocyclopropane carboxylic acid oxidase (*ACO*), ET-response factor (*ERF1*)), and ABA (*NCED2*) was evaluated by RT-qPCR in treated leaves before and after pathogen infection (Figure 2).

The expression of *ERF1*, *PR1* and *PR3c* was upregulated in leaf tissues at 2 dpt with TAG and IFP48 compared to control (Figure 2A). The expression of *PR1* increased by approximately 5- and 13-fold, respectively. Those of *ERF1* and *PR3c* reached 3 and 4-fold, while *PR2* was slightly upregulated by IFP48. However, no consistent changes were observed in the expression of *ACO*, *LOX9* and *NCED2* in response to TAG and IFP48.

The expression of *PR-1* was increased after *P. viticola* inoculation compared to mock inoculated leaves, while it was downregulated after *B. cinerea* inoculation (Figure 2B). The *PR1* expression remained elevated in IFP48-treated plants following *P. viticola* inoculation, but it downregulated after *B. cinerea* inoculation, compared to the inoculated mocks. TAG had only slight effect on the *PR1* level, especially after *B. cinerea* challenge. The expression of *PR2* increased more following infection with *B. cinerea* than with *P. viticola*. Interestingly, *PR2* expression slightly increased in IFP48-treated leaves following *P. viticola* inoculation, but remained constant after *B. cinerea* inoculation. The pathogen infection did not affect *ERF1* expression, except in TAG- and IFP48-treated leaves inoculated with *B. cinerea*. Following *P. viticola* inoculation, the expression of *PR-3c* remained low, while it increased after *B. cinerea* inoculation. However, compared to the inoculated mocks, IFP48 treatment resulted in an enhanced expression of *PR3c* upon *P. viticola* inoculation, while *PR3c* expression remained constant after *B. cinerea* inoculation. No significant change was observed at the transcriptional level of *NCED2* involved in ABA synthesis, in response to both IFP48 and TAG (Figure 2A), and even after pathogen inoculation (Figure 2B).

### 2.3. IFP48 and TAG Induce Slight Changes in the Expression of Sugar Transport-Related Genes

Considering the importance of sugar resources in the outcome of the plant-pathogen interaction, we investigated the possible contribution of sugar transport and cleavage processes in grapevine response to TAG and IFP48 treatment before and after pathogen infection. We monitored the expression of two genes which encode high affinity hexose transporters of the plasma membrane (VvHT1 and VvHT3), two genes encoding sugar transporters of the SWEET (Sugar Will be Eventually Exported Transporters) family (Sweet2a and Sweet4), and GIN1 gene which encodes vacuolar invertase 1. Data showed that both VvHT1 and VvHT3 did not show any significant transcriptional regulation after TAG and IFP48 treatment (Figure 2C) and after *P. viticola* or *B. cinerea* inoculation (Figure 2D). The expression of Sweet2a was not affected by both treatments, while that of Sweet4 was upregulated at 2 dpt with both IFP48 and TAG (Figure 2C). At 24 hpi with *B. cinerea*, the expression level of Sweet4 remained low as in water-treated plants (Figure 2D). However, after infection with *P. viticola*, Sweet4 was upregulated. The TAG-treated leaves showed a low expression of Sweet4 after pathogen inoculation, while in IFP48-treated leaves the expression of Sweet4 remained high following *P. viticola* infection, but not *B. cinerea* (Figure 2D). The expression of vacuolar invertase gene GIN1 decreased in response to IFP48, and more drastically following TAG treatment, compared to control (Figure 2C). However, after pathogen challenge GIN expression did not show any relevant modification among treatments (Figure 2D).

### 2.4. IFP48 and TAG Enhance Stilbene Phytoalexin Accumulation

IFP48 exhibited a stronger effect on the biosynthesis of stilbenes than TAG (Figure 3). This was especially noticeable at 2 dpt, in case of resveratrol (Figure 3A). The amount of resveratrol reached approximately 5.5 µg per gram fresh weight, which was 46 folds higher than control. At 2 dpt, the amount of resveratrol was also slightly induced by TAG, reaching approximately 3.5 folds compared to control (Figure 3A). After *P. viticola* inoculation, the amount of resveratrol was slightly induced in both IFP48- and to a lesser extent in TAG-treated plants at both 1 dpi and 2 dpi (Figure 3B). A slight increase of resveratrol content was also observed after *B. cinerea* inoculation in IFP48-treated leaves (Figure 3C).

Both IFP48 and TAG had almost the same effect on the amount of ε-viniferin (a resveratrol dehydrodimer) as on resveratrol before and after leaf inoculation with *P. viticola*. The amount of ε-viniferin peaked at 2 dpt in response to IFP48, reaching approximately 3 µg per gram fresh (15 folds higher than control treatment) (Figure 3D). As for resveratrol, the early induction of ε-viniferin was followed by a sudden drop at 3 dpt and 4 dpt, while keeping the statistical significance over the control. Following *P. viticola* inoculation, ε-viniferin was slightly higher in IFP48-treated plants compared to water control (Figure 3E). After *B. cinerea* inoculation, a slight increase of ε-viniferin amount was detected at 1 dpi, but not at 2 dpi (Figure 3F).

The content of δ-viniferin increased in IFP48- and TAG-treated plants compared to control before pathogen inoculation (Figure 3G). δ-viniferin was accumulated later compared to resveratrol and ε-viniferin. At 3 and 4 dpt, the amount of δ-viniferin increased in response to IFP48 compared to water and TAG treatments. After pathogen inoculation, the pattern of δ-viniferin was very similar to that of ε-viniferin (Figure 3H,I). The amount of δ-viniferin was higher in IFP48-treated leaves at 2 dpi with *P. viticola*, but no significant change was observed in TAG-treated plants (Figure 3H). In both TAG and IFP48-treated leaves, the level of δ-viniferin was comparable among treatments after infection with *B. cinerea* (Figure 3I).

The amount of piceid, a glycosylated form of resveratrol, increased over time of treatment with TAG and IFP48 (Figure 3J). The level of piceid remained high after pathogen infection and it was higher in IFP48-treated plants compared to water-treated plants at 1 dpi with *P. viticola* (Figure 3K). However, no significant differences were observed between treatments in the amount of piceid after *B. cinerea* inoculation (Figure 3L).

### 2.5. IFP48 and TAG Induce Differential Change in Tagatose and Endogenous Sugar Amounts in Leaf Tissues before and after P. viticola Challenge

The modulation of sugar content in plants is considered to function as a signal to prime immune response. In this study we investigated whether the level of tagatose (derived from IFP48 and TAG) accumulated inside the leaf tissues and the changes in endogenous sugars in association to plant resistance against *P. viticola*. Grapevine leaves accumulated a high and similar amount of D-tagatose at 2 dpt with IFP48 and TAG, reaching approximately 1800 μg/g FW, while control leaves contained less than 1 μg/g FW (Figure 4A). However, at 3 dpt the accumulated D-tagatose level decreased by about 80% in IFP48-treated leaves and by 95% in TAG-treated leaves, compared to D-tagatose quantified at 2 dpt. After *P. viticola* inoculation, the amount of D-tagatose was higher in IFP48-treated plants compared to control plants (Figure 4B).

Plants treated with IFP48 and TAG did not show any significant change in the amounts of endogenous D-glucose, D-ribose, D-fructose, D-mannose and D-galactose (Figure 4C). The amounts of D-maltose and D-trehalose were reduced at 2 dpt in IFP48- and TAG-treated plants compared to control plants. After *P. viticola* inoculation, the amounts of D-glucose, D-fructose and D-maltose decreased, while those of D-galactose and D-trehalose increased, compared to mock plants (Figure 4D). No changes were observed in the levels of D-ribose and D-mannose after pathogen infection. The amounts of D-glucose, D-fructose and D-maltose remained constant in both TAG and IFP48-treated plants after *P. viticola* inoculation, compared to control-infected plants (Figure 4D). However, D-galactose and D-trehalose contents were reduced by both treatments after pathogen inoculation. The amount of D-trehalose was reduced by approximately 3- and 2-fold in IFP48- and TAG-treated plants in comparison to water-treated plants, respectively (Figure 4D).

## 3. Discussion

### 3.1. IFP48 Induces Grapevine Resistance against P. viticola, but Not B. cinerea, by Modulating the Expression SA, JA/ET-Responsive Defense Genes and Potentiating Phytoalexin Accumulation

Rare sugars have been shown to reduce severity of various plant diseases and their effectiveness depends on the pathogen lifestyle [25,28,29,30,31]. For instance, D-tagatose was more effective against biotroph oomycetes including grapevine downy mildew, than necrotrophs like *B. cinerea* [31,32,33,34]. However, molecular mechanisms underlying the efficiency of D-tagatose or D-tagatose-based products against grapevine diseases are not understood. The broader effectiveness of D-tagatose was shown to be linked to its direct action on the pathogen, rather than its capacity to activate plant defense mechanisms [31,32]. In this study, we showed that foliar applications of TAG-based product IFP48 and pure TAG reduce *P. viticola* infection, but not *B. cinerea*, in the susceptible cv. Chardonnay under greenhouse conditions. The TAG-based product (IFP48) was more effective than the TAG itself, suggesting a potential contribution of co-formulants, which would also be related to the bioavailability or the fate of D-tagatose in plant tissues. Given that grapevine leaves were Tween80-washed before pathogen inoculation, it can be inferred that protective effects triggered by IFP48 and TAG might rely at least in part on the induced grapevine resistance.

Our data provided evidence that the induced protection against *P. viticola* is associated to the IFP48′s and to a lesser extent TAG’s ability to induce a sweet immune response (before pathogen infection) and to potentiate the activation of various defense-related genes and stilbene phytoalexin accumulation upon pathogen challenge. It is noteworthy that most of defense responses remained highly upregulated in IFP48-treated plants after infection with *P. viticola*, but inconsistent following challenge with *B. cinerea*. The induced resistance to *P. viticola* is correlated to a strong activation of SA- and JA/ET-dependent defense pathways by IFP48 and to a lesser extent by TAG before *P. viticola* inoculation, as emphasized by an upregulation of *PR1* and *PR2* (markers of SA), *ERF1* and *PR3c* genes (markers of JA/ET), and stilbene phytoalexin accumulation in the absence of pathogen infection. In most cases, some SA-dependent defenses and viniferin accumulation were highly upregulated in response to IFP48 compared to TAG treatment, and even maintained to some extent after *P. viticola*, but not *B. cinerea* infection. This indicates that IFP48-mediated resistance against *P. viticola* is at least to some extent dependent on the ability of IFP48 to elicit grapevine defenses, but also to prime the plants for enhanced SA-dependent immune reactions upon *P. viticola* challenge. Data are in agreement with previous research showing that enhanced resistance to downy mildew involves SA- and to some extent JA/ET-responsive immune defenses, such as in the case for glucan-based elicitors [3,5,8,44] or β-amino-butyric acid (BABA) [45]. The interplay between SA and JA/ET signaling pathways seems to play a key role in the effective resistance against *P. viticola* [46]. This is in line with the TAG-induced weaker resistance compared to IFP48, since despite a priming effect of TAG for enhanced expression of *PR1*, the expression of JA/ET responsive genes was not affected. Another assumption would be that TAG could be easily used, in absence of appropriate co-formulant, as a nutrient source by phyllospheric microorganisms as reported in the greenhouse [35], or by the pathogen itself, thus reducing its efficiency against the disease.

It is likely that the ineffectiveness of IFP48 or TAG against *B. cinerea* could also be due to weak induction of JA/ET pathways or to the ability of *B. cinerea* to overcome SA-dependent defenses. Our data are, at least in part in agreement with other reports [36] showing that foliar treatment of tomato with D-allose primed plants for enhanced expression of *PR-1* (SA-pathway), leading to a slight reduction of the gray mold symptoms (by approximately 15%). Although resistance to necrotrophs is generally associated to JA/ET pathways [1,13], which were activated by IFP48 and to a lesser extent by TAG, no significant protection against *B. cinerea* was conferred by both treatments. It has been reported that D-allose treatment of riceinduced systemic acquired resistance (SAR)-like necrotic symptoms and the upregulation of *PR-1* gene [29]. Such plant responses are frequently related to hypersensitive reaction (HR) [44,47], which could even facilitate plant infection by the necrotrophic pathogen like *B. cinerea* [4,48].

Data also showed that *NCED2* gene, involved in ABA biosynthesis, did not respond to TAG or IFP48 treatments either before or after pathogen inoculation. This suggests that TAG- or IFP48-activated immune response is independent on ABA. These results are in contrast to the D-allose effect in rice, which induced a strong and transient accumulation of ABA and upregulation of ABA biosynthetic genes, including *NCED2* [49]. It is likely that D-tagatose uses a different route than D-allose retarding defense induction. Nevertheless, it seems that *NCED2* is rather developmentally regulated, while *NCED1* might be linked to ABA biosynthesis under stress conditions [50]. Thus, further research is needed to understand whether ABA can be involved in the induced resistance against *P. viticola*.

Increasing the synthesis of stilbenes is among the arsenal of defense response implemented by grapevine plants against various pathogens, including *B. cinerea* and *P. viticola* [6,51,52]. IFP48 and TAG induced a significant accumulation of stilbenes, which remained high even after pathogen inoculation. This was especially important in the case of resveratrol dimers, trans-*ε*-viniferin and trans-*δ*-viniferin, which are known to display antifungal activity and reduce the release and mobility of *P. viticola* zoospores [51,52,53,54]. However, upon *B. cinerea* challenge only resveratrol content remained high in the IFP48-treated plants, while trans-piceid amount was unchanged. These results are in line with other research [54] showing that resveratrol and piceid have little or no toxic activity against *P. viticola*, whereas viniferins are highly toxic and can be considered important markers of grapevine resistance to downy mildew.

### 3.2. IFP48 and TAG Induce Changes in Sugar-Related Gene Expression and Sugar Amounts

IFP48 and TAG treatments did not induce any consistent change in the expression of genes encoding hexose transporters (*VvHT1* and *VvHT3*), even after pathogen inoculation. Among the two *VvSweet* (Sugar Will be Eventually Exported Transporters) genes tested, only *VvSweet4* was slightly upregulated at 2 dpt with IFP48 (and to a lesser extent with TAG) before the pathogen challenge. The enhanced expression of *VvSweet4* was correlated with the increased level of D-tagatose inside the leaf tissues. It has been reported that the expression of *VvSweet4* in grapevine hairy roots mediates increase in sugar levels and facilitates the expression of phenylpropanoid-related genes [55]. Thus, it becomes tempting to assume that the upregulation of *VvSweet4* could play a significant role in sugar homeostasis into the cells, thereby contributing to IFP48-induced resistance against *P. viticola*. *VvSweet4* could also be involved in the accumulation of exogenous D-tagatose into the cells, counteracting the infection structures of *P. viticola* [31]. *VvGIN1*, encoding a vacuolar invertase, was downregulated by IFP48 and to the lesser extent by TAG, especially after pathogen inoculation. The *VvGIN1* regulation seems to correlate with the observed influx of D-tagatose, as abundant hexose, into leaf tissues. Plant invertases are known to hydrolyze sucrose into glucose and fructose moieties [56,57], thus shifting apoplastic sucrose/hexose ratio in favor of hexoses. It is suggested that the newly accumulated D-tagatose, due to IFP48 or TAG treatment, could reduce the expression *VvGIN1*.

Some differences between IFP48 and TAG treatments were also observed regarding the amount of D-tagatose accumulated in leaf tissues, which remained slightly higher in IFP48-treated leaves (3.5-fold) compared to the TAG treatment. This is consistent with the efficiency of IFP48 in inducing resistance against *P. viticola*. Similar results have been reported in a recent study [26], showing higher level (3 to 4-fold) of D-tagatose in cucumber leaves following root treatment with IFP48 compared to pure TAG. This indicates that co-formulant in IFP48 can facilitate D-tagatose influx into leaf tissues or protected it from the consumption by phyllospheric microbes [35], thereby improving grapevine resistance against *P. viticola*. Although co-formulant included in IFP48 is protected by industrial secret and cannot be tested separately, the characterization of IFP48 clarified the mechanism of action of the formulated D-tagatose against *P. viticola*. Further experiments with different D-tagatose formulations are necessary to understand the exact contribution of co-formulants in D-tagatose efficacy against *P. viticola*.

IFP48 and TAG did not induce major changes in sugar content, except the amounts of D-maltose and D-trehalose which showed a significant decrease. This suggests possible triggered sugar-specific singling events resulting in the induction of immune responses. The slight reduction of glucose and fructose content in inoculated leaves of the IFP48- and TAG-treated plants could be linked to the observed downregulation of *VvGIN*. A direct connection between D-tagatose and D-glucose metabolism has been reported in the oomycete *P. infestans* [27]. D-glucose may also be redirected to the phenylpropanoid pathway for the synthesis of phytoalexins, resulting in enhanced resistance [58]. Structural similarity of D-tagatose with common sugars is probably a crucial factor contributing to its effect. D-fructose (a D-tagatose epimer) have been shown to hinder the efficacy of D-tagatose against *P. infestans* [27], *Streptococcus mutans* [59], and grapevine downy mildew [34]. This suggests that D-tagatose can interact with the fructose metabolism in an antagonistic manner. However, IFP48 and TAG did not induce any significant change in the amount of mannose and ribose, probably due to their relative structural dissimilarity. Yet, D-tagatose was reported to interfere with D-mannose metabolism in *Hyaloperonospora arabidopsidis* [31] and reduce the amount of D-mannose in *P. infestans* [27], leading to the inhibition of both oomycetes. The slight reduction of D-maltose might indicate possible impairment of starch breakdown or to enhanced transglucosidase activity, using maltose as a donor to transfer one glucose moiety to a polysaccharide [60]. The amounts of galactose (monosaccharide) and trehalose (a non-reducing disaccharide composed of two glucose units) were significantly reduced in IFP48- and TAG-treated grapevine plants and after *P. viticola* infection. Although trehalose is considered as an important signal that regulates defense response [61], it may also affect the biology of the pathogen by upregulating the expression of virulence genes, or by activating its metabolism and promoting its development within the host [62]. Galactose may also contribute to the biosynthesis of pectin in the plant primary cell wall [63]. This suggests that the reduction of trehalose and galactose in IFP48-treated plants could contribute to the reduction of downy mildew disease.

In conclusion, foliar application of a D-tagatose-based product can increase grapevine resistance to *P. viticola*, but not to *B. cinerea*. IFP48 can enhance SA- and to some extent JA/ET-dependent immune responses and viniferin accumulation in grapevine leaves upon *P. viticola* infection. IFP48 also improves the accumulation of D-tagatose in the leaf tissues, which seems to interact with endogenous sugar contents, thereby enhancing its eliciting defense activity. Further studies with different D-tagatose formulations are necessary to clarify the contribution of co-formulants in D-tagatose efficacy.

## 4. Materials and Methods

### 4.1. Plant Material and Growth Conditions

Cuttings were collected from ten-year-old grapevine plants (*Vitis vinifera* L., cv. Chardonnay) and placed in the cold chamber at 4 °C. After a two-month period, cuttings were surface-sterilized for 4 h with 0.05% cryptonol (8-hydroxyquinoline sulfate), washed with distilled water and right away placed back in the cold chamber for another week in darkness. Afterwards, cuttings were hydrated in the distilled water bath for 16 h at 25 °C and potted in the 5 L pot (three cuttings per pot) containing horticultural soil (Sorexto M4600, Grenoble, France). Pots were placed in a greenhouse (25 °C day/night, 60% relative humidity, and 16 h photoperiod using natural daylight or artificial lamps). Plants were watered with tap water twice per week and grown for 8 weeks.

### 4.2. Preparation of Rare Sugar and Treatment

D-tagatose-based product (IFP48, wettable powder containing 80% D-tagatose w/w; Lot: 17F-5786) and pure D-tagatose (TAG) were provided by Bi-PA (Biological Products for Agriculture, Londerzeel, Belgium). IFP48 and TAG were diluted in sterile ultra-pure water, filtered through syringe filters with sterile membranes (0.45 µm pore size) (VWR International) and prepared at 5 g/L as an optimal concentration. Plants were treated with IFP48, TAG and water (control) by using the hand sprayer until the homogenous coverage of axial and abaxial leaf sides was reached. Plants were left in the greenhouse for two additional days after treatment before pathogen inoculation.

### 4.3. Pathogen Growth Conditions and Inoculum Preparation

*Plasmopara viticola* isolate was purified as a fresh single spore from infected leaves of the susceptible Chardonnay in vitro plantlets [64]. Pathogen was maintained on the leaves of *V. vinifera* cv. Chardonnay cuttings and subcultured every week for the fresh inoculum supply [44]. To obtain sporangia for the inoculum, leaves with oil spot symptoms were put in the dew chamber (100% RH, 25 °C) overnight. After the sporulation was promoted, lesions were washed with sterile distilled water and the concentration of sporangia in the suspension was adjusted to 1 × 10^5^ with Malassez hemocytometer.

*Botrytis cinerea* strain 630 was grown on potato dextrose agar (PDA) medium for 14 days at 22 °C under continuous light for sporulation. Conidia were scratched and suspended in 10 mL sterile water as described in Aziz et al. [44]. To eliminate the mycelium, the obtained suspension was filtered through sterile filter paper. Then, concentration of conidia was measured with Malassez hemocytometer and adjusted to 1 × 10^6^ conidia/mL.

### 4.4. Pathogen Inoculation

At two days post-treatment (dpt) with IFP48, TAG and water, 2 leaves per plant (3rd and 4th leaves from the top) were detached and washed three times with 0.001% Tween80 to remove external IFP48 and TAG, which could directly affect the pathogen growth. Leaves were then placed in glass Petri dishes on wet Whatman paper and inoculated in two different ways: (i) For the purposes of disease severity assessment, leaves were inoculated by placing five 20-µL drops of a fresh suspension of *P. viticola* (1 × 10^5^ sporangia/mL) and three 5-µL drops of conidial suspension of *B. cinerea* (1 × 10^6^ conidia/mL) on their abaxial side. (ii) For the purposes of analyzing the expression of targeted defense responses and sugar accumulation inside the leaf tissues, abaxial side of the leaves was inoculated by spraying 1 × 10^5^ sporangia/mL or 1 × 10^6^ conidia/mL of fresh suspensions of *P. viticola* and *B. cinerea*, respectively, using the hand sprayer. Mock was inoculated by spraying distilled water.

### 4.5. Disease Severity Assessment

In case of *B. cinerea*, disease severity was measured at 5 and 14 dpi by measuring the diameter of necrotic lesions as described by Aziz et al. [44]. In case of *P. viticola*, disease severity was determined by measuring sporangial density by using leaf discs of 18-mm diameter (10 leaf discs per plant). At 7 dpi leaf discs were randomly pooled in the groups of 5 leaf discs (each group consisted of 5 leaf discs from different plants that underwent the same treatment), placed in a falcon tube 50 mL, and covered with 1 mL sterile distilled water. After 1 h shaking time, suspended sporangia were counted under the light microscope using a Malassez hemocytometer. Each *B. cinerea* and *P. viticola* experiment was performed with 12 leaves (derived from 6 plants) per treatment, and experiments were repeated three times.

### 4.6. RNA Extraction and Analysis of Gene Expression by RT-qPCR

Sampling was done at zero-, 2-dpt with IFP48, TAG and water, and at 1 dpi and 2 dpi with *P. viticola*, *B. cinerea* or mock inoculation. Total RNA was extracted from 50 mg of leaf powder using PlantRNA (Invitrogen, Thermo Fisher Scientific, Waltham, MA, USA). Complementary DNA was synthesized from 1 µg of total RNA. Reverse transcription was done with Verso cDNA Synthesis Kit (Thermo Fisher Scientific, Inc., Waltham, MA, USA) according to the manufacturer’s instructions. Using CFX 96TM Real Time System (Bio-Rad, Marnes-la-Coquette, France) and Absolute qPCR Mix, SYBR Green, ROX (Thermo Fisher Scientific), with qPCR, we assessed the expression profiles of genes markers of SA pathway including pathogenesis related protein-1 (*PR-1*) and 1,3-glucanase (*PR-2*) [48]; JA pathway: 9-lipoxygenase (*LOX9*) [45] and acidic class IV chitinase (*PR-3*) [48,65]; ET: 1-aminocyclopropane carboxylic acid oxidase (*ACO*) and Ethylene response factor 1 (*ERF1*) (transcription factor in both JA/ET signaling) [48]; ABA: 9-cis-epoxycarotenoid dioxygenase 2 (*NCED2*); genes encoding for vacuolar invertase *VvGIN1* and sugar transporters such as *VvHT1, VvHT3, VvSweet2a and VvSweet4*. The specific primers listed in Appendix A were designed by Primer 3.0 software (Applied Biosystems) [48]. PCR reactions were set at 95 °C for 10 s (denaturation) and 60 °C for 45 s (annealing/extension) for 42 cycles as described in Lakkis et al. [64]. After the evaluation with Bio-Rad CFX MANAGER software v.3.0, that expression of Elongation factor 1-alpha (*EFα1*) and 60S ribosomal protein (*60SRP*) remained unchanged in all tested conditions, therefore, they were selected as reference genes. Transcript levels of target genes were quantified using the standard curve method. Normalization was done against *EFα1* and *60SRP* as internal controls. Water control at 0 hpt was considered as reference sample (1× expression level). For each experimental condition, PCR reactions were performed in duplicate. Twelve leaves from 6 plants were used for each condition. Three independent experiments were performed.

### 4.7. Phytoalexins Extraction and Analysis

Leaf samples were collected at zero and 2-dpt with IFP48, TAG and water, and at 1 dpi and 2 dpi with *P. viticola*, *B. cinerea* or mock inoculation. Stilbene phytoalexins were extracted from 200 mg of leaf powder prepared in liquid Nitrogen. In the first step, 2 mL of methanol-to-water 85 % *v*/*v* was added to each tube containing the plant material. Samples were shaken at 800 rpm for 2 h in darkness at the room temperature, then centrifuged for 15 min at 15,000 rpm and 4 °C. Supernatants were collected, protected from the light and stored in the fridge. For the second step, pellets were resuspended in 1 mL of pure methanol, shaken at 800 rpm for 1 h in the darkness, and centrifuged for 15 min (15,000 rpm and 4 °C). Supernatants were collected and pooled together with supernatants obtained in the first step, then dried with the speed vacuum at 45 °C (Speed-Vac, Eppendorf France SAS, Montesson, France). Residues were re-solubilized with 1 mL of pure methanol, filtered through 0.22 µm PTFE filters and stored into 2 mL amber vials for UPLC analysis. Trans-piceid, resveratrols, ε- and δ-viniferins were analysed using ACQUITY™ UPLC system (Waters Corporation, USA) with Acquity^TM^ UPLC BEH C18 1.7 µm 2.1×100 mm column heated at 40 °C. Water and acetonitrile with 0.1% phosphoric acid were used to elute stilbenes at the flow rate of 0.5 mL min^–1^ over 7 min [64]. Detection of phytoalexins was performed with an Acquity fluorimeter (Waters Corporation, Milford, MA 01757, USA) with an excitation/emission wavelength of 330/375 nm and phytoalexins were quantified with reference to retention time and calibrated with external standards. Twelve leaves from 6 plants were used for each condition, and experiments were repeated three times.

### 4.8. Sugar Quantification

Leaf samples were collected at 2 dpt with IFP48, TAG and water before infection and after one day of *P. viticola* or mock inoculation. Leaves were collected from each plant, washed three times with 0.001% Tween80 to remove external IFP48 and TAG and immediately frozen in liquid nitrogen for subsequent quantification of sugars inside the leaf tissues. The amount of D-tagatose, D-glucose, D-ribose, D-fructose, D-mannose, D-galactose, D-maltose and D-trehalose in ground leaf samples was assessed by ion chromatography [66]. Briefly, samples were dissolved in ultrapure water, filtered through a 0.45 µm PTFE membrane (Sartorius, Goettingen, Germany) and analysed with an ionic chromatograph ICS 5000 (Dionex-Thermo Scientific, Waltham, MA, USA), equipped with an autosampler, a quaternary gradient pump, a column oven and a pulsed amperometric detector with a gold working electrode and a palladium counter electrode. The separation was obtained by injecting 5 µL of diluted sample onto a CarboPac PA200 3 × 250 mm analytical column (Dionex-Thermo Scientific, Waltham, MA, USA), preceded by a CarboPac PA200 3 × 50 mm guard column (Dionex-Thermo Scientific), with a KOH gradient (from 1 to 100 mM) at 0.4 mL/min flow rate. Sugar content was expressed in µg per g fresh weight leaf material by using a calibration curve of each pure sugar (Sigma-Aldrich, Merc, Kenilworth, NJ, USA) dissolved in ultrapure water within a range between 0.2 and 40 µg/mL.

### 4.9. Statistical Analysis

Quantification of *P. viticola* sporangia was repeated three times. Data were analyzed with R software version 3.6.0, and One-way ANOVA statistical test with pairwise comparisons and Student’s t-test were used to detect significant differences (*p* ≤ 0.05). In case of disease severity caused by *B. cinerea*, quantification of sugars in leaf tissues, phytoalexin and gene expression analyses, experiments were carried out three times. Data analysis was performed with R software version 3.6.0. One-way ANOVA statistical test was used with SPSS 20 software with post-hoc Tukey’s test HSD (Honestly Significant Difference) to detect significant differences (*p* ≤ 0.05) between treatments.

## Figures and Tables

**Figure 1 plants-11-00296-f001:**
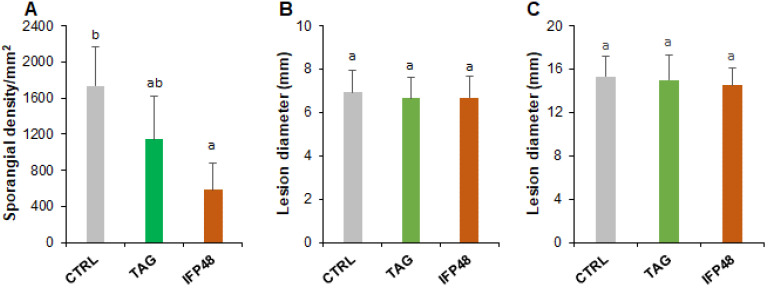
Downy mildew and gray mold diseases in grapevine leaves after treatment with IFP48 and TAG. Plants were sprayed with IFP48 or TAG at 5 g/L or water as control (CTRL). After two days of treatment (dpt), leaves were detached, washed and inoculated with *P. viticola* (**A**) or *B. cinerea* (**B**,**C**), then disease was evaluated by counting the *P. viticola* sporangia at 7 dpi (**A**), and by measuring the *B. cinerea* necrosis size at 5 dpi (**B**) and 14 dpi (**C**). Data are means ± SD from three independent experiments and different letters indicate significant differences according to the One-way ANOVA and Tukey’s test (*p* ≤ 0.05).

**Figure 2 plants-11-00296-f002:**
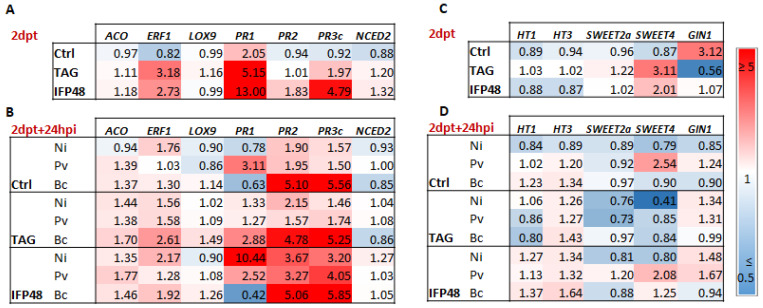
Relative expression of defense- and sugar-related genes in grapevine leaves after treatment with IFP48 and TAG (**A**,**C**) and inoculation with *P. viticola* and *B. cinerea* (**B**,**D**). Plants were treated with IFP48 and TAG at 5 g/L or water (control, Ctrl), 2 days post treatment (dpt) samples were inoculated with *P. viticola*, *B. cinerea* or mock (non-infected, Ni) and collected before (**A**,**C**) and after 24 h of inoculation (2 dpt + 24 hpi) (**B**,**D**). Data are means from three independent experiments. Ctrl, control; Ni, non-infected; Pv, *P. viticola*; Bc, *B. cinerea*. Heatmaps represent changes in transcript expression levels as indicated by the color legend.

**Figure 3 plants-11-00296-f003:**
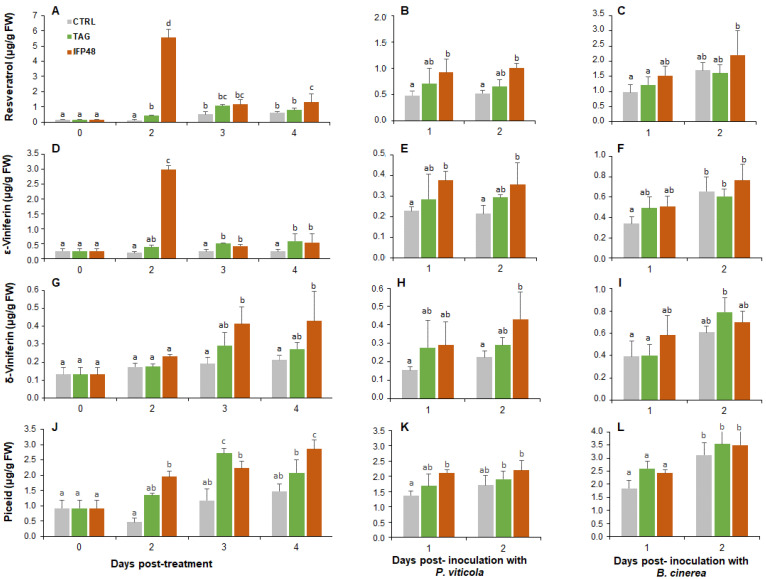
Stilbene accumulation in grapevine leaves after treatment with IFP48 and TAG and inoculation with *P. viticola* and *B. cinerea*. Plants were treated during four days with IFP48 and TAG at 5 g/L, or water (**A**,**D**,**G**,**J**), or treated for 2 days then infected with *P. viticola* (**B**,**E**,**H**,**K**) or *B. cinerea* (**C**,**F**,**I**,**L**). Data are means from three independent experiments and different letters indicate significant differences according to the One-way ANOVA and Tukey’s test (*p* ≤ 0.05).

**Figure 4 plants-11-00296-f004:**
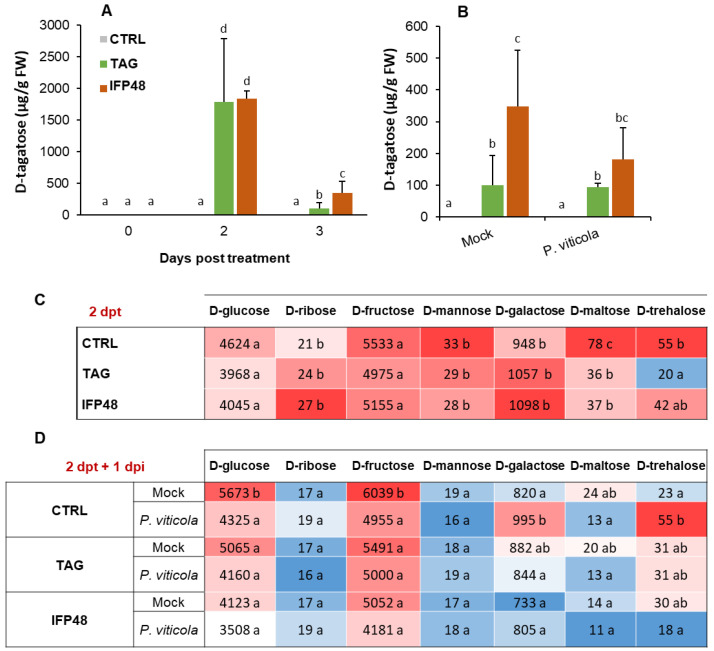
Amounts of D-tagatose and endogenous sugars in grapevine leaf tissues after treatment with IFP48 and TAG and inoculation with *P. viticola*. Plants were treated during three days with IFP48 and TAG at 5 g/L, or water (**A**,**C**), or treated for 2 days then infected with *P. viticola* (**B**,**D**). D-tagatose (**A**,**B**), D-glucose, D-ribose, D-fructose, D-mannose, D-galactose, D-maltose and D-trehalose (**C**,**D**) were quantified. Data are means from three independent experiments and different letters indicate significant differences according to the One-way ANOVA and Tukey’s test (*p* ≤ 0.05).

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
