# Peer review of "D-Tagatose-Based Product Triggers Sweet Immunity and Resistance of Grapevine to Downy Mildew, but Not to Gray Mold Disease"

_plants, 2022, doi:10.3390/plants11030296_

Round 1

Reviewer 1 Report

The authors investigated the induced resistance of grapevine leaves against Plasmopara viticola and Botrytis cinerea by a rare sugar-based product (IFP48) and its active ingredient D-tagatose (TAG), in order to elucidate molecular mechanism involved in defense-related metabolic regulations before and after pathogen challenge. The article is well written and organized, the work is correct realized, and results are conclusive and support the discussions and conclusions. Please see my few comments below.  

As a general observation, many references are quite old. I recommend to the authors is possible to replace them with newer, let’s say published in the last 10 years.

It would be nice if the authors can highlight the originality of the paper at the end of introduction.

Figure’s 1 capture is very long. Please write a short relevant caption and discuss the Figure in the body text. More concrete, a significant part of the caption can be transferred in the text.

Figure’s 2 capture – the same comment as before. Please keep the caption concise.

Figure’s 3 & Figure’s 4 captions – same as before.

Section 4.1 – with what water were the cuttings washed and which water were they watered? Was tap water?

Author Response

We thank the reviewer for his instructive comments and suggestions. We have taken all comments into account and made all necessary corrections.

As a general observation, many references are quite old. I recommend to the authors is possible to replace them with newer, let’s say published in the last 10 years. The list of references has been reduced by choosing the most relevant and recent in the field

It would be nice if the authors can highlight the originality of the paper at the end of introduction. It was done (see L114-118)

Figure’s 1 capture is very long. Please write a short relevant caption and discuss the Figure in the body text. More concrete, a significant part of the caption can be transferred in the text. The legends are reduced and the details are given in the section Materials and methods

Figure’s 2 capture – the same comment as before. Please keep the caption concise. It was done

Figure’s 3 & Figure’s 4 captions – same as before. It was done

Section 4.1 – with what water were the cuttings washed and which water were they watered? Was tap water? It was specified

Reviewer 2 Report

The article, no doubt, is of some interest, but there are a number of comments. In Fig. 1A, the standard deviations are too large, so the differences between the Tukey options are unlikely to be significant, and therefore do not correspond to the specified p ≤ 0.05. At the same time, in Fig. 1 A, the data are given in absolute values, while in Fig. 1 B and C – data are given in relative (%) values. Obviously, uniformity in the presentation of data should be maintained. In addition, calculations of root-mean-square deviations are possible only for experimental options relative to control ones, taken as 100%. In this case, why in the control, where everything is taken as 100%, ±errors are also indicated.

In my opinion, there is an excessive amount of cited literature in the article - 97 sources are more suitable for a literary review. There is no need to provide 3-4 links to support any fact.

Author Response

We thank the reviewer for his instructive comments and suggestions. We have taken all comments into account and made all necessary corrections.

The article, no doubt, is of some interest, but there are a number of comments. In Fig. 1A, the standard deviations are too large, so the differences between the Tukey options are unlikely to be significant, and therefore do not correspond to the specified p ≤ 0.05. At the same time, in Fig. 1 A, the data are given in absolute values, while in Fig. 1 B and C – data are given in relative (%) values. Obviously, uniformity in the presentation of data should be maintained. In addition, calculations of root-mean-square deviations are possible only for experimental options relative to control ones, taken as 100%. In this case, why in the control, where everything is taken as 100%, ±errors are also indicated.

The data for Figure 1A were reviewed and corrected based on a high number of samples. Similarly, the presentation of data in Figure 1B and 1C were also given in absolute values and standardized as Figure 1A.

In my opinion, there is an excessive amount of cited literature in the article - 97 sources are more suitable for a literary review. There is no need to provide 3-4 links to support any fact. The list of references has been reduced by choosing the most relevant and recent in the field

Reviewer 3 Report

The author of the manuscript entitled "-tagatose-based product triggers sweet immunity and resistance of grapevine to downy mildew, but not to grey mold disease" need some minor revision prior to its final acceptance.

Comments

Title: I think “but not to gray mold disease” is not necessary for the title.

Introduction:

Line 37: Replace “Vitis vinifera” with “Vitis vinifera L.”

Result

2.1. Remove  “but not B. cinerea”

Line 221: Typo error

“After pthogen inoculation,”

Line 513: Correction required

“Leaf samples were collected at zero-,”

Author Response

We thank the reviewer for the instructive comments and suggestions. We have taken all comments into account and made all necessary corrections.

Title: I think “but not to gray mold disease” is not necessary for the title.

We chose to keep "gray mold" in the title, because this work focuses also on the comparison of TAG and IFP48 efficacy against the two pathogens and explains why IFP48 is effective against the oomycete P. viticola, but less effective against the fungus B. cinerea. Thus, this is an important message in the paper and we hope the Reviewer will appreciate the value of this information.

Introduction:

Line 37: Replace “Vitis vinifera” with “Vitis vinifera L.”. it was done

Result

2.1. Remove  “but not B. cinerea” same explanation as for the title.

Line 221: Typo error

“After pthogen inoculation,” it was corrected

Line 513: Correction required

“Leaf samples were collected at zero-,” it was corrected

Reviewer 4 Report

Reviewer report for paper entitled “D-tagatose-based product triggers sweet immunity and resistance of grapevine to downy mildew, but not to gray mold disease”.

This work shows a novel idea of using rare sugars and D-tagatose to induce plant immunity (sweet immunity) against grape disease (downy mildew). This new approach of biological control using rare sugar is of high interest for both scientific and industrial society. The novelty of work is clear and authors studied in depth the mechanism of plan immune enhancement in gene level for enhancement of bioactive ingredients against plant pathogens. This work is close to a recently published work by some of the authors of this work which studied also in depth the effect of Tagatose to enhance the cucumber immunity against powdery mildew https://doi.org/10.1016/j.cropro.2021.105753

The work is of high interest but needs some minor revision before acceptance as follows:

Keywords: recommend to remove (rare sugar) and add IFP48.

Introduction: this part needs to be improved and focus on plant immune response and effect of rare sugars (D-tagatose) on fungal pathogens resistance and plant physiology. The number of references in this part need to be reduced, having 67 references in introduction part is too much.

Results part: This part is well written and used a well-designed approach which is suitable for this type of research and the results are well presented. It was also very interesting to study in depth the concentration of sugar content after plant treatment and how sugar concentrations affected based on rare tagatose treatment.

Discussion part: this pare is well written and designed and discussed clearly the results and demonstrate clearly the mechanism of the enhancement of plant resistance against fungal pathogens.

Materials and Methods: This part is well written and provided suitable references for the methods used.

Conclusion part is missing and authors need to provide conclusion at the end of this work to highlight the outcomes, novelty and suggestions for future work in this field.

In general, this work is very interesting and provided a novel research and I recommend for publication after minor revision.

Author Response

We thank the reviewer for the instructive comments and suggestions. We have taken all comments into account and made all necessary corrections.

Keywords: recommend to remove (rare sugar) and add IFP48. It wa done

Introduction: this part needs to be improved and focus on plant immune response and effect of rare sugars (D-tagatose) on fungal pathogens resistance and plant physiology. The number of references in this part need to be reduced, having 67 references in introduction part is too much. It was improved and references have been selected and the number was reduced

Results part: This part is well written and used a well-designed approach which is suitable for this type of research and the results are well presented. It was also very interesting to study in depth the concentration of sugar content after plant treatment and how sugar concentrations affected based on rare tagatose treatment. Thank you!

Discussion part: this pare is well written and designed and discussed clearly the results and demonstrate clearly the mechanism of the enhancement of plant resistance against fungal pathogens. Thank you!

Materials and Methods: This part is well written and provided suitable references for the methods used. Thank you!

Conclusion part is missing and authors need to provide conclusion at the end of this work to highlight the outcomes, novelty and suggestions for future work in this field.

A conclusion was provided at the end of the discussion (L404-410) with some relevant suggestions for future studies

In general, this work is very interesting and provided a novel research and I recommend for publication after minor revision.

Thank you!